# Solid matrix-assisted printing for three-dimensional structuring of a viscoelastic medium surface

Sungchul Shin[1], Hojung Kwak[1], Donghyeok Shin [1] & Jinho Hyun[1,2]*

*Gluconacetobacter xylinus* (*G. xylinus*) metabolism is activated by oxygen, which makes the formation of an air-medium interface critical. Here we report solid matrix-assisted 3D printing (SMAP) of an incubation medium surface and the 3D fabrication of bacterial cellulose (BC) hydrogels by in situ biosynthesis of *G. xylinus*. A printing matrix of polytetrafluoroethylene (PTFE) microparticles and a hydrogel ink containing an incubation medium, bacteria, and cellulose nanofibers (CNFs) are used in the SMAP process. The hydrogel ink can be printed in the solid matrix with control over the topology and dimensional stability. Furthermore, bioactive bacteria produce BC hydrogels at the surface of the medium due to the permeability of oxygen through the PTFE microparticle layer. The flexibility of the design is verified by fabricating complex 3D structures that were not reported previously. The resulting tubular BC structures suggest future biomedical applications, such as artificial blood vessels and engineered vascular tissue scaffolding. The fabrication of a versatile free-form structure of BC has been challenged due to restricted oxygen supplies at the medium and the dimensional instability of hydrogel printing. SMAP is a solution to the problem of fabricating free-form biopolymer structures, providing both printability and design diversity.

[1] Department of Biosystems and Biomaterials Science and Engineering, Seoul National University, Seoul 08826, Republic of Korea. [2] Research Institute of Agriculture and Life Sciences, Seoul National University, Seoul 08826, Republic of Korea. *email: jhyun@snu.ac.kr

Bacterial cellulose (BC) is a polysaccharide synthesized from many different microorganism strains and a hydrogel of a complex networked structure in which nanometer-sized cellulose fibers are intertwined[1]. BC has a unique set of properties, including high mechanical strength, water content, crystallinity, and purity[2–7]. Moreover, because it causes no specific immune response and exhibits high biocompatibility, BC-based materials are promising candidates for a wide variety of biomedical applications, including tissue scaffolding, artificial blood vessels, and skin substitutes[8–16]. In spite of these features, a technology that allows for flexible designs and controllable three-dimensional (3D) structuring of BC hydrogels remains necessary before such applications can be implemented. The restricted oxygen supply has also proven a major hurdle to overcome. Because BC is biosynthesized at the surface of oxygen-rich media, controlling the features of the air–liquid interface is also essential for practical 3D structuring[17,18].

Viscous inks containing biocompatible rheology modifiers have been used to form a variety of features on substrates via direct 3D printing[19–21]. However, fabricating a free-form structure is not practical; this is because printed gel structures are easily deformed by gravity, and contact with substrates flattens the features. In addition, the substrates are not permeable to oxygen, which is essential for aerobic biosynthesis. Recently, a 3D structure of biopolymer was prepared by combining the aeration protocols that allow the medium surface to be controlled[22]. Complex 3D structures of BC with a resolution of about 50 μm were prepared in a mold with hydrophobic particles[22]. Higher resolutions of the BC could be fabricated in patterned superhydrophobic–hydrophilic surfaces and at the interface of a soft-lithographic polydimethyl siloxane mold[23,24]. Because the limited choice of needles for direct 3D printing, the resolution of solid matrix-assisted 3D printing (SMAP) using a viscous ink is comparatively low. However, SMAP enables control over the topology and interconnectivity, which is a feature that is achieved uniquely by 3D printing.

SMAP begins with the combined material system of a viscoelastic hydrogel ink and a hydrophobic fluidic solid matrix. SMAP uses a cellulose nanofiber (CNF)-based viscoelastic ink containing active bacteria printable in a hydrophobic solid particle matrix. Support for extruded hydrogel structures is achieved by printing the ink in fluidic matrices, such as a viscose liquid or non-adhesive solid particles. The fluidity of the selected solid particles enables 3D printing of hydrogel ink by rapidly filling the space created by needle movement. The hydrophobic solid matrix provides dimensional stability for the viscoelastic 3D structures as well as a gap space between the particles through which the oxygen required for biosynthesis of cellulose can be delivered. To fabricate free-form BC structures, the printed feature is exposed to air in a three-dimensional manner while retaining the 3D structure.

To date, an oxygen-permeable silicon mold has been used to control the shape of BC structures, but modification of feature structures has proven difficult. Fabricating a 3D structure of BC in solid-phase incubation using SMAP of CNF hydrogel ink containing bacteria proved challenging, and no reports of fabrication through conventional 3D printing and solution incubation have been published. In contrast, SMAP allows for flexible design of more complex structures, including spheres, tubes, coils, and connected rings. Especially, BC tubular structures can be used as engineered tissue support for blood vessels or neural regeneration. SMAP can also be applied to structure hydrogel inks containing various types of aerobic microorganisms. In addition, it is expected that SMAP will be expanded by diversifying the solid particles constituting the matrix such as metals, ceramics, and wood materials, which would upgrade the conventional 3D printing technology further.

## Results

**SMAP of bacteria containing ink**. Fabrication of free-form BC structures was performed by SMAP, as shown in Fig. 1, beginning with the selection of a solid matrix for the effective and controlled creation of 3D hydrogel structures. First, the solid matrix should be sufficiently hydrophobic and immiscible with hydrogel-based printing ink. Second, the space made by needle movements should be refilled by solid matrix materials immediately after the needle passes through. Third, air should be able to reach the printed hydrogel features through the microparticles for biosynthesis of cellulose at the interface. A spherical polytetrafluoroethylene (PTFE) microparticle was chosen as the material for the solid matrix due to its hydrophobicity and bioinertness (Fig. 1a). Water repellency against the PTFE surface facilitated the formation of air–hydrogel interfaces and the space between the PTFE microparticles permitted the supply of oxygen needed for the biosynthesis of cellulose (Fig. 1b).

Bacterial bioactivity was restricted to near the surface region due to the limited supply of oxygen to the 3D hydrogel structures. Oxygen levels were higher at the surface of the 3D printed hydrogel facing the solid matrix and lower toward the center of the printed structures (Fig. 1c). The difference in oxygen levels resulted in heterogeneous biosynthesis of BC, and a tubular structure of BC could be fabricated by incubating the CNF hydrogel printed in a filament shape for a controlled incubation time. SMAP enabled controlled 3D fabrication of hydrogel-containing bacteria. Successfully constructed complex structures included coil, connected ring, tetrahedron, stacked lattice, and sandglass structures, which have not been reported in 3D printing to date (Fig. 1d).

**Rheological characterization of CNF hydrogel ink containing bacteria and the control of printability**. The physical stability of printed hydrogel structures was affected significantly by the rheological properties of inks as well as the hydrophobicity of the solid matrix. Figure 2a shows the change in viscosity according to the amount of CNF added to the ink. As CNF content increased, the viscosity of the ink increased due to entanglement between the CNFs. The viscosity of the ink was effectively increased with a small amount of CNFs. The CNF hydrogel-based ink showed a shear-thinning behavior in which viscosity gradually decreased over a wide range of shear rates. In effect, viscosity was lowered by the shear force acting at the wall surface of the needle, allowing CNF hydrogel ink to discharge easily from the needle. Oscillatory shear stress sweep showed that the CNF hydrogel ink was predominantly elastic at a high CNF concentrations and viscous at low CNF concentrations (Fig. 2b). Above the shear yield stress, the shear elastic modulus ($G'$) dropped dramatically with increased shear stress, allowing the ink to be printed at modest pressures. The storage modulus and the viscosity of the CNF hydrogel ink decreased sharply when the applied shear stress was raised from 0.5 to 300 Pa at intervals of 60 s (Fig. 2c). The ink could be ejected easily through the needle, and stable 3D structures could be formed after ink discharge.

The PTFE microparticles attached uniformly to the printed ink surface, forming a discrete interface without spreading along the PTFE microparticle surfaces (Fig. 2d). Different 3D structures of CNF hydrogel were routinely printed inside the PTFE solid matrix. However, due to the viscoelastic property of CNF hydrogel, the printed 3D structure was deformable by inter- and intra-structure stresses. Dimensional deformation was evaluated by the diameter change of the printed CNF hydrogel filaments under ultraviolet illumination (Fig. 2e). At a CNF content of 0–0.75%, the printed ink filaments coagulated into separate drops and showed a larger diameter change, while at a

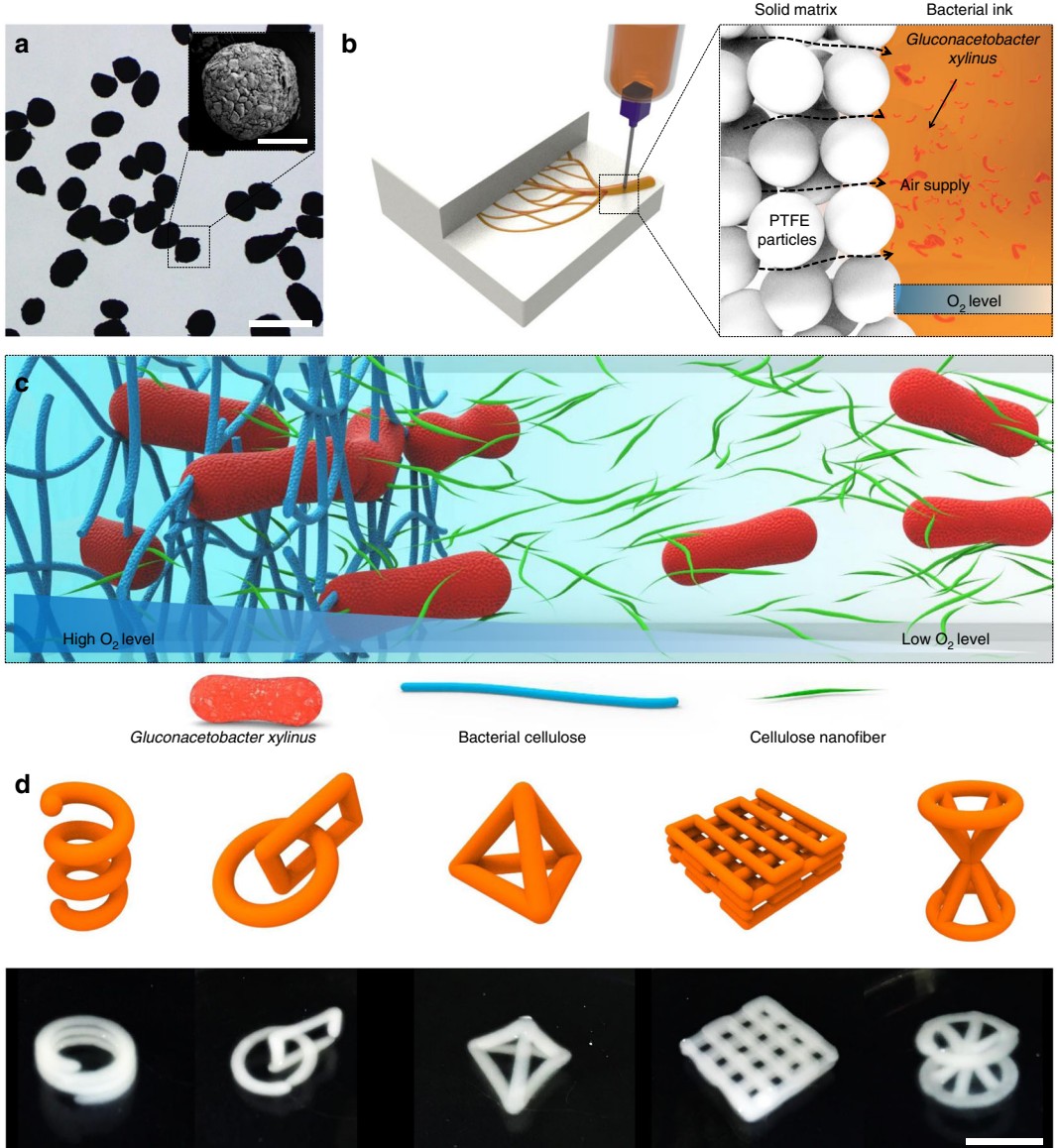

**Fig. 1** Solid matrix-assisted 3D printing of bacteria containing ink. **a** Optical and electron microscopic images of PTFE microparticles used as a solid matrix. **b** Bacteria containing ink were printed inside of solid matrix. Oxygen was supplied through the solid particles to the surface of the CNF ink containing *G. xylinus*, allowing the bacteria to metabolize. **c** BC was produced at the surface of the printed CNF hydrogel ink in high oxygen conditions. **d** Coil, connected ring, tetrahedron, stacked lattice, and sandglass structure of CNF/BC after incubation as printed according to the CAD designs. Scale bars are 500 μm in the optical microscopic image and 100 μm in the FE-SEM image of **a**, and 1 cm in the optical microscopic images of **d**

CNF content of 1.0–1.5% for the printed structures could be maintained without significant changes in the diameter (Fig. 2e).

The capillary, elastic, and interfacial adhesion forces should be balanced to prevent deformation of the printed structures in the matrix. It was possible for the hydrophilic CNF ink to deform into separate round-shaped drops in the hydrophobic PTFE matrix rather than retain the printed continuous structures due to its higher surface tension. Such stress can be represented by the Laplace pressure ($\Delta P$), and the yield stress ($\sigma_Y$) above the pressure could be found by adjusting the CNF content in the ink. The yield stress of the CNF hydrogels increased with CNF content (Fig. 2f). Interfacial adhesion stress ($K_R$) of the solid particles with the ink also resisted the Laplace pressure. The value of $K_R$ was not changed by the addition of CNF to the ink up to 0.25%, but changed significantly with the addition of 1.25% CNF (Fig. 2g). The ink at different CNF contents was printed in the solid matrix and the printing stability increased with the CNF content, implying a $\Delta P < \sigma_Y + K_R$. For ink with a CNF content of 0.25%, the yield stress was 0.15 Pa and the interfacial adhesion stress was 15 Pa, which was not high enough to maintain the printed spherical shape of separate drops. As the CNF content in the ink increased, the yield stress and interfacial adhesion stress of the printed CNF hydrogel also increased. For ink with a CNF content of 1.25%, the yield stress was 13 Pa and the interfacial adhesion stress was 56 Pa, which was high enough to retain printed CNF hydrogel structures (Fig. 2f, g).

Features between 500 and 2500 μm in diameter were printed by varying the translation speed at the three different ink flow rates. The printed feature size was inversely related to the speed of the needle, but showed similar feature sizes at the flow rates used in the research. The linear correlation of the feature size and the translation speed enabled controlled structuring of the hydrogel ink and enhanced printing performance (Fig. 2h).

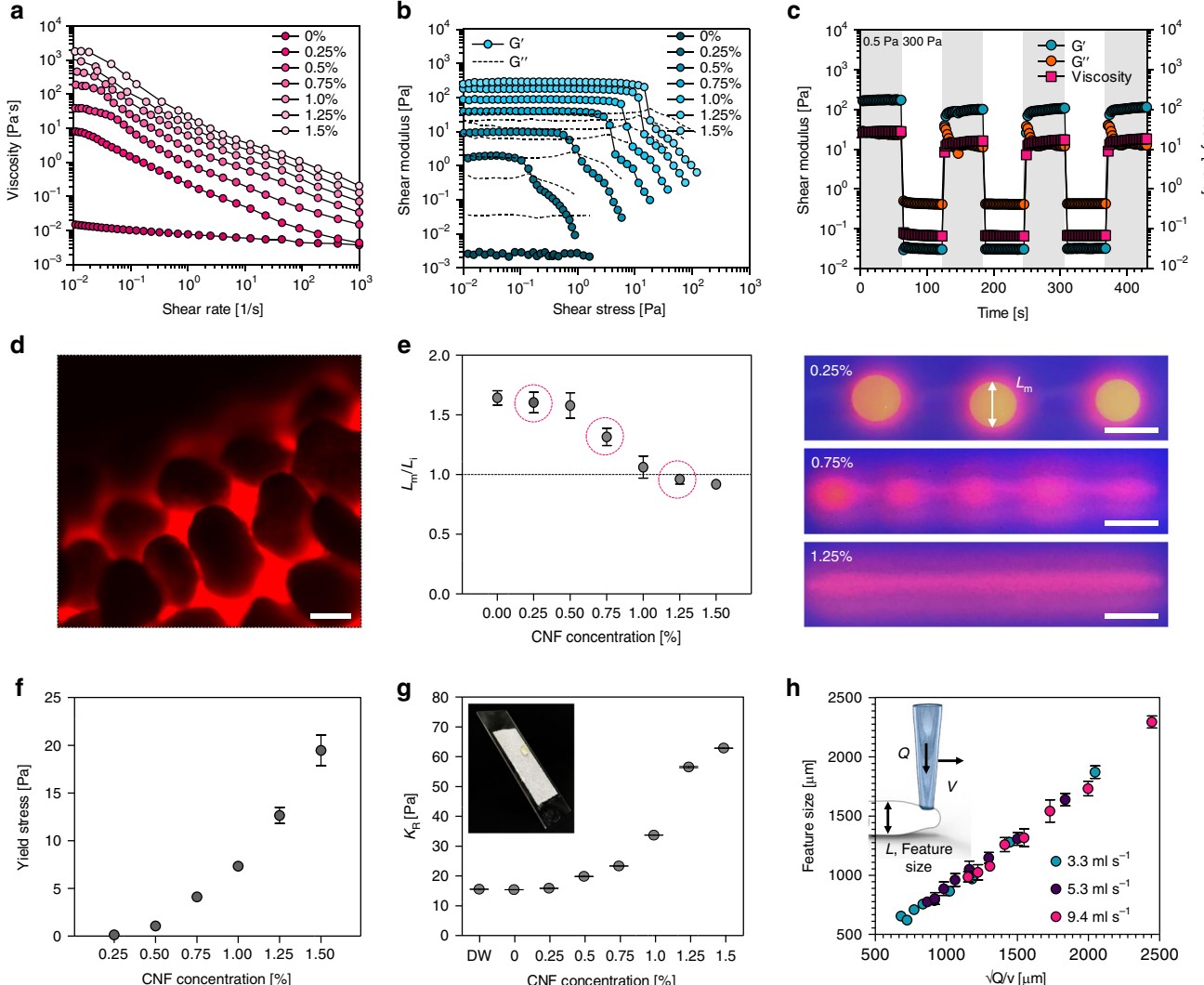

**Fig. 2** Rheological characterization of CNF hydrogel ink containing bacteria and the control of printability. **a** Viscosity of inks at different contents of CNF as a function of the shear rate. **b** Storage and loss moduli of inks at different contents of CNF in stress sweeps. **c** Cyclic stress time sweep for 1 wt% CNF hydrogel ink. For cyclic stress, the shaded regions were at a low stress of 0.5 Pa and unshaded regions were at a high stress of 300 Pa. **d** Fluorescent microscope image of PTFE particles bound at the surface of the printed CNF hydrogel structure. **e** Ratio between $L_m$ (measured diameter of printed CNF hydrogels) and $L_i$ (intended diameter of printed CNF hydrogels) at different CNF contents. Ink with high printing stability showed a ratio close to 1. **f** Change of yield stress according to CNF content. **g** The change of interfacial adhesion stress at the surface of PTFE microparticles facing CNF hydrogel according to CNF content. **h** The feature size of printed objects could be controlled by the tangential velocity of the needle ($v$) and the flow rate of the ink through the needle ($Q$). The printed feature size showed a linear increase following an ideal behavior across a wide range of velocities and flow rates at 1.25 wt% CNF content. Scale bars are 100 μm in **d** and 5 mm in **e**. Data are presented as mean ± s.d. ($n = 3$ in **e**, **f**, and **g**, $n = 9$ in **h**)

**Characterization of the solid matrix and control of the printing features**. For bacterial bioactivity in the printed structures, an air-permeable solid matrix is required. Air permeability was evaluated by Gurley air resistance, the time taken for 100 mL of air to pass through the matrix. PTFE microparticles with a diameter of 100 μm had a Gurley air resistance of 7 s per 100 mL, which was lower than the value of Grade 2 filter paper (26.6 s per 100 mL), implying acceptable air permeability (Fig. 3a and Supplementary Fig. 1). Air permeability increased with the size of the PTFE microparticles of the solid matrix; those 660 μm in diameter exhibited a Gurley air resistance of <1 s per 100 mL.

Fabrication of free-form structures in a viscoelastic matrix required precise control of rheological properties of the matrix[25–30]. The fluidity of the solid particles in the matrix determined the printability because the space developed by the needle movement should be filled with the neighboring matrix particles as the needles

moves. When the needle moved in the PTFE solid matrix, a crevice was formed along the needle's path and the filling rate of the crevice differed according to particle size. Particle fluidity was investigated by measuring the angle of repose with a different size of PTFE microparticle. The angle of repose gradually increased as particle size decreased (Fig. 3b). This can be attributed to denser packing of particles as particle size decreased.

Shape deformation was closely related to the formation of crevices and the filling of this space with ink. Figure 3c shows the length ratio of the x-axis and y-axis with the cross-sectional observation of the printed line. As the particle size shrank, the length ratio increased and the cross-sectional shape of the printed CNF hydrogel ink gradually changed from circular to elliptical. The fluidity of the smaller particles decreased and the crevice space made by the needle movement was filled with ink until it was covered with neighboring particles. A solid matrix with an average

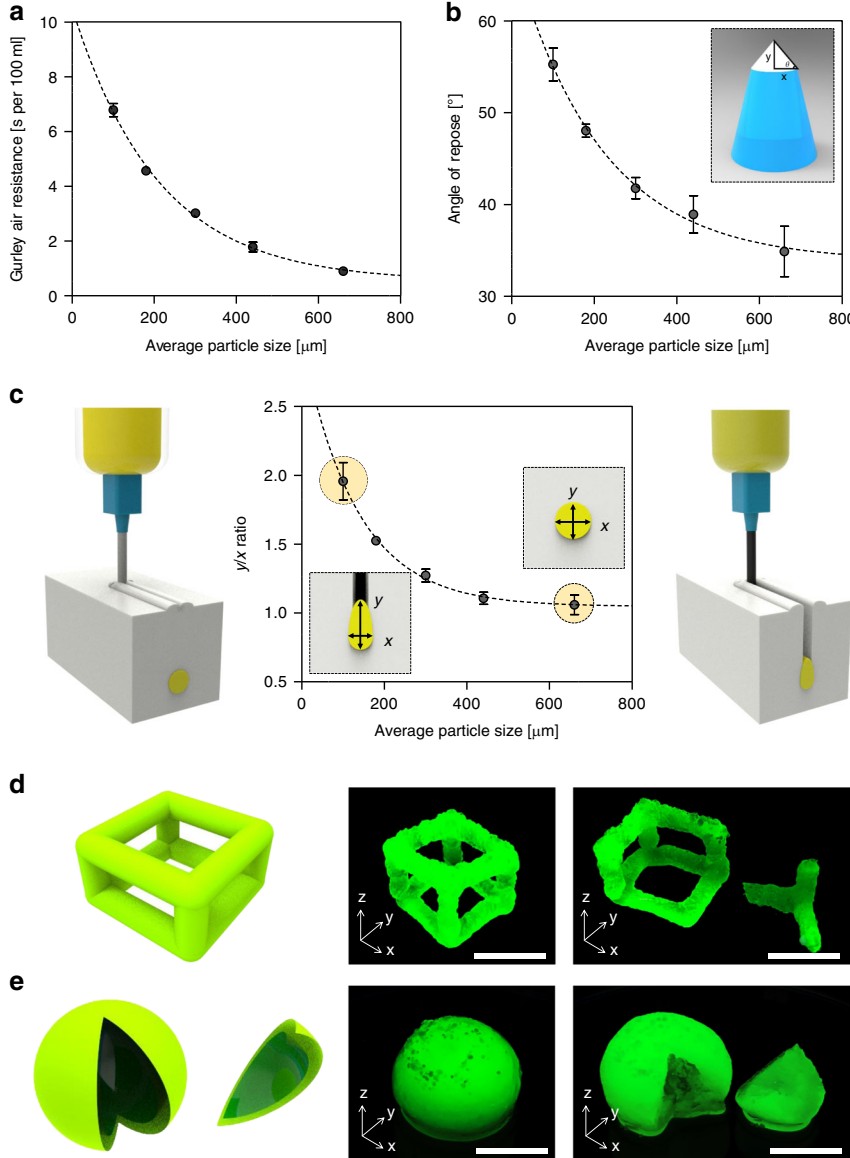

**Fig. 3** Characterization of the solid matrix and control of the printing features. **a** Gurley air permeability of PTFE microparticle matrix according to average particle size. **b** Angle of repose according to average particle size. **c** The ratio between the width and height of printed lines using different particle sizes of solid matrix. Creation of cuboid (**d**) and hollow sphere (**e**) structures with CNF hydrogel ink containing green fluorescent microspheres. Those structures were imaged under UV illumination. Scale bars are 1 cm in **d**, **e**. Data are presented as mean ± s.d. (*n* = 5 in **a**, **b**; *n* = 3 in **c**)

particle size of 100 μm produced a printed feature *y/x* ratio of close to 2 (highly elliptical). The shape deviation of printed features could be prevented by filling the crevice space with the neighboring PTFE microparticles immediately after the needle was moved. The filling rate was determined by the fluidity of the particles. The larger particles filled the crevice faster and enabled fabrication of structures with an ideal *y/x* ratio close to 1 (circular).

Cuboid and spherical structures were printed with hydrogel ink containing 1.25% CNF and *Gluconacetobacter xylinus* (*G. xylinus*) using SMAP of 660 μm PTFE (Fig. 3d, e). Dimensional stability was confirmed by incorporating green fluorescent particles with CNF ink for visualization under ultraviolet illumination. They were then incubated as printed in an incubation chamber for 7 days to produce BC. BC was biosynthesized at the surface of the printed features, forming thick tubular structures. The hollow structures were achieved by removing the CNF hydrogel ink with DW.

**BC network properties depending on the growing media conditions**. The network structure of BC changes depending on the growing media conditions[31,32]. The yield, fiber diameter, and mesh size of BC were analyzed by varying the concentration of mannitol, which was a carbon source contained in the ink, to 1.25, 2.5, 5, and 10%. SMAP was performed by positioning the tip of the nozzle in the solid matrix of 1 cm depth from the surface (Fig. 4a). Printing was carried out by fixing the extruding volume to 200 μL to form a spherical structure. The yield of BC production was evaluated by measuring the dry weight after the incubation up to 10 days. As the concentration of mannitol increased, the yield of BC increased. A measure of 1.25% of mannitol concentration showed a plateau in the increase of BC dry weight. The other mannitol concentrations showed continuous increase of dry weight, but the significant difference of BC yield was not observed at the concentration over 2.5% (Fig. 4b). The network structure of BC obtained after the incubation for 7 days showed the average

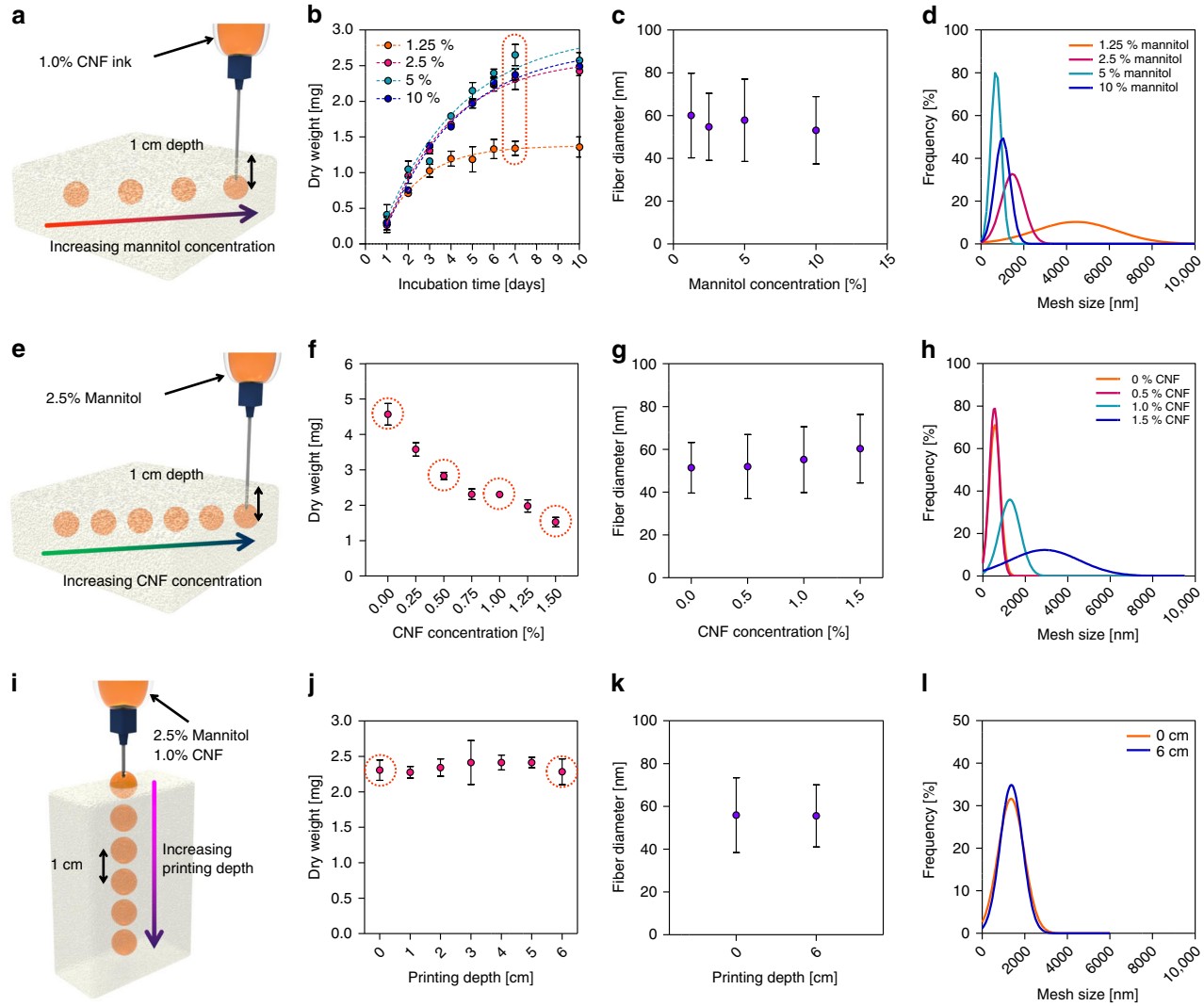

**Fig. 4** BC network properties depending on the growing media conditions. **a** Schematic illustration of SMAP of droplets at different mannitol concentrations. **b** The yield of BC according to mannitol concentrations. Average fiber diameter (**c**) and mesh size (**d**) of BC according to mannitol concentration. **e** Schematic illustration of SMAP for droplets at different CNF concentrations. **f** The yield of BC according to CNF concentration after incubating for 7 days. Average fiber diameter (**g**) and mesh size (**h**) of BC according to CNF concentration. **i** Schematic illustration of SMAP for droplets at different printing depth. **j** The yield of BC according to printing depth after incubating for 7 days. Average fiber diameter (**k**) and mesh size (**l**) of BC according to the printing depth. Data are presented as mean ± s.d. ($n = 3$ in **b**, **f**, and **j**)

diameter of fiber about 55 nm regardless of the concentration of mannitol (Fig. 4c). The distance between the junction points of the BC fibers was defined as the mesh size[33,34]. Mesh size decreased and the size distribution became narrower as the concentration of mannitol increased (Fig. 4d).

The viscosity of the ink containing bacteria is critical in the yield and network structure of BC due to the restriction of bacteria locomotion in the media. Under the fixed mannitol concentration of 2.5%, the concentration of CNF was varied up to 1.5%. SMAP was performed as described above to form a spherical structure in the solid matrix (Fig. 4e). As the concentration of CNF increased from 0 to 1.5%, the yield of BC gradually decreased with increasing CNF concentration (Fig. 4f). This was due to the reduced mobility of the bacteria at viscose CNF ink because the restricted locomotion of bacteria reduced the bioactivity for the synthesis of BC[19]. The fiber diameters did not show significant differences with respect to the concentration of CNF, but the difference in mesh size was significant (Fig. 4g, h).

The production of BC in hydrogels may significantly be influenced by the air-feeding levels. It is critical to confirm even BC production with respect to the printing depth for the fabrication of homogeneous BC hydrogel structures. SMAP of CNF ink containing bacteria was performed at different depths of the solid matrix up to 6 cm from the matrix surface (Fig. 4i). The yield of BC exhibited an almost constant amount of BC up to 6 cm deep, inferring that the oxygen levels were similar regardless of depth in the experiment (Fig. 4j). In addition, the diameters of the BC fibers were similar between the near matrix surface and the 6 cm deep bulk region. Interestingly, the mesh sizes were also similar for different printing depths, which differs from the results obtained for different medium conditions (Fig. 4k, l).

Possible changes in the network structure of the BC according to the printed structure were investigated using two extreme shapes of BC (Supplementary Fig. 2a). A straight line and the curved edge of BC structure produced at the same printing depth were characterized, and no significant differences in fiber diameter or mesh size of the BC were observed, implying the

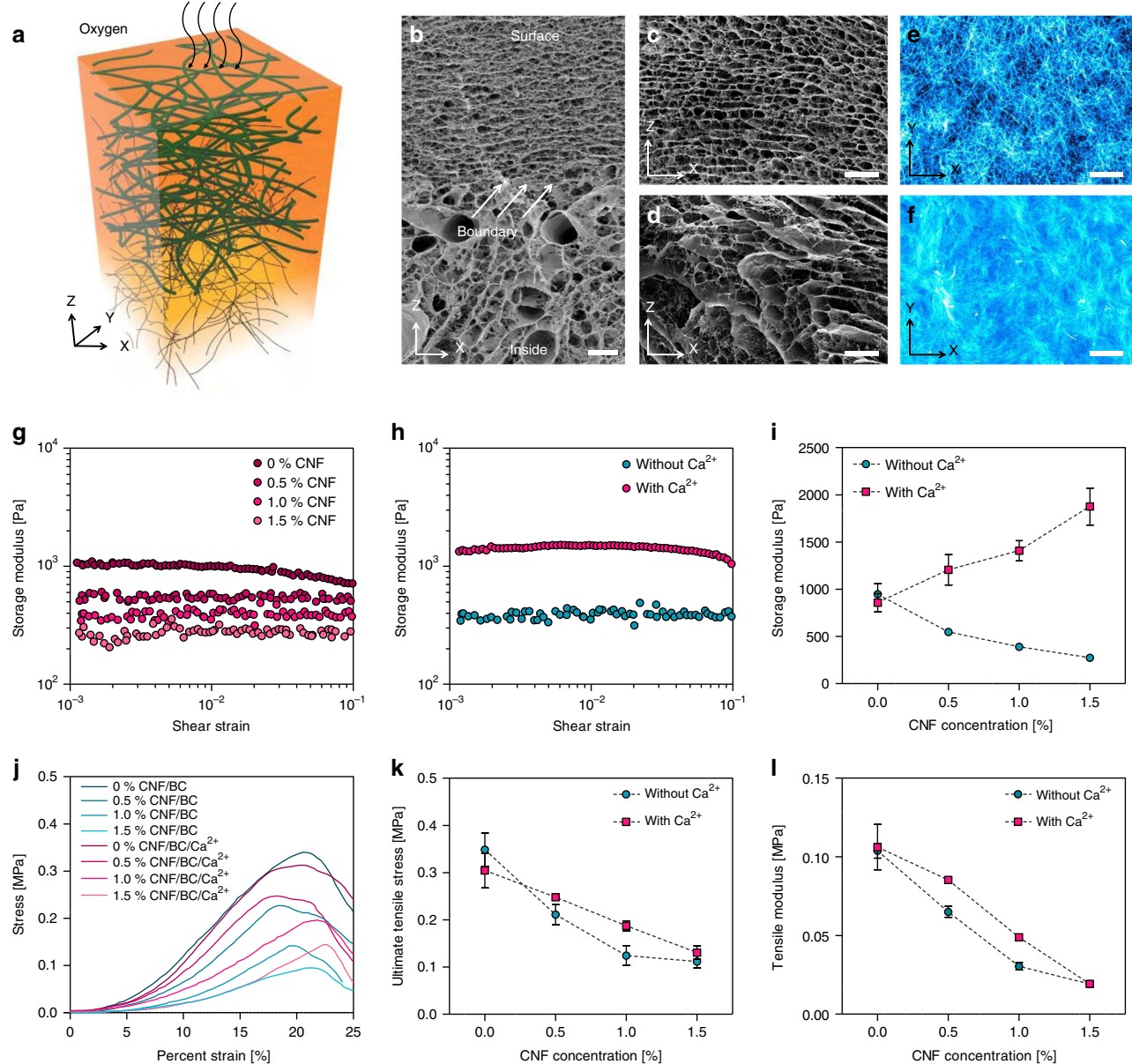

**Fig. 5** Physical properties of the CNF/BC hydrogel. **a** Schematic representation of BC biosynthesized in CNF hydrogel with a culture medium. Cross-sectional FE-SEM images of CNF/BC tubes (**b**), multilayered BC at the surface region (**c**), and the compositing morphology of CNF and BC at the deeper region (**d**). Cellulose fibers at the surface region (**e**) and the deeper region (**f**) were observed by staining with Calcofluor white. Shear strain dependency of storage modulus $G'$ for CNF/BC hydrogels at different CNF contents from 0 to 1.5% (**g**) and calcium ion treatment (**h**). **i** Changes in storage modulus by CNF content and calcium ion treatment. **j** Representative tensile stress–strain curve of CNF/BC hydrogels at different CNF contents and calcium ion treatments. **k** Maximum tensile stress of CNF/BC hydrogels. **l** Tensile elastic modulus of CNF/BC hydrogels at 10% strain. The scale bars are 20 μm in **b–f**. Data are presented as mean ± s.d. ($n = 3$ in **l**, **k**, and **l**).

homogeneous BC hydrogels exhibited morphological diversity (Supplementary Fig. 2b, c).

**Physical properties of the CNF/BC hydrogel**. The morphological structure of the BC produced at the surface of the printed CNF hydrogel showed a typical multiple-layer structure similar to BC produced in a static culture (Fig. 5a). Cross-sectional images showed two main compartments: a highly layered region of BC and a porous composite region consisting of BC and CNFs (Fig. 5b). Bacteria tended to move to the surface for active biosynthesis and form thin cellulose layers with regular spacing between them. Because the BC layers were closely networked by ultrathin cellulose fibers, the structure retained its original morphology in the

freeze-drying process (Fig. 5c). Meanwhile, CNFs in the hydrogel formed a distinct morphology—they were dehydrated and aggregated into dense plates due to the formation of ice particles in the freeze-drying process. The space between ice particles remained as a large pore after drying, but the BC fibers retained the inherent networked structures connecting the CNF plates (Fig. 5d). The CNF was highly nanofibrillated and the diameter of CNF was about 18.6 nm, as measured with atomic force microscopy, which was much smaller than the diameter of BC fibers of about 55 nm; this is because the CNFs did not have a tight junction between the fibers, unlike BC fibers (Supplementary Fig. 3).

BC fibers biosynthesized with the printed ink structures were stained with Calcofluor white for the specific labeling of cellulose.

The networked BC fiber structure could be clearly seen at the surface, and the BC networks were intercalated with CNFs in the deeper region, forming a dense composite structure (Fig. 5e, f).

In general, BC hydrogels have a low storage modulus and are easily deformed by external stress. The presence of CNF did not ensure dimensional stability of the CNF/BC hydrogel structures in spite of intercalation with BC nanofibers. CNF/BC hydrogels did not retain their shape and were easily bent by gravity without ionic crosslinking. CNFs intercalated with BC enabled the formation of ionic crosslinking due to the presence of carboxymethyl groups that supplied $Ca^{2+}$ ions (Supplementary Fig. 4). This improved the dimensional stability of BC significantly. CNFs dispersed homogeneously and filled the space between BC fiber networks, and the $Ca^{2+}$ ions induced strong bonding between CNFs, preventing deformation of the CNF/BC composite structures (Supplementary Fig. 4).

The storage modulus of CNF/BC composites is shown in Fig. 5g. Without $Ca^{2+}$ ionic crosslinking, the storage modulus of the CNF/BC composites gradually decreased as the content of CNF in the printing ink increased due to the electrostatic repulsion between CNFs and lower network density of BC (Fig. 5h, i). In contrast, ionic crosslinking between the CNFs increased the storage modulus of the composites significantly as the CNF content increased (Fig. 5h, i). Frequency sweep tests are widely used to obtain information regarding the stability of three-dimensional cross-linked networks. After the strain-sweep test of $Ca^{2+}$ ion-treated CNF/BC composites, the condition for the frequency sweeps was selected at 0.5% strain to ensure the linear viscoelastic range during the test (Supplementary Fig. 5a). $Ca^{2+}$ ion-treated CNF/BC composites was subjected to a frequency sweep from 0.1 to 10 Hz. $G'$ value exhibited a plateau in the range 0.1–10 Hz, which was indicative of a stable network hydrogel. In addition, a temperature sweep test was performed to determine the susceptibility of hydrogel structure to the change of temperature. $G'$ and $G''$ values of $Ca^{2+}$ ion-treated CNF/BC composites were constant in the temperature range from 4 °C to 80 °C, confirming the high thermal stability of the hydrogel (Supplementary Fig. 5b). Meanwhile, the ultimate tensile stress of composites showed an increased value after ionic crosslinking with $Ca^{2+}$ ions (Fig. 5j, k), but the tensile modulus decreased with the increase of CNF content differently from the shear modulus (Fig. 5l). Ionic crosslinking appeared to increase the packing density thickness, but promoted the localization of CNFs with regard to length, forming loosely bound BC fibers.

**Preparation of a hollow cellulose tube**. A hollow tubular structure of BC was prepared by biosynthesizing cellulose at the surface of CNF hydrogel filaments containing *G. xylinus* (Fig. 6a). *Gluconacetobacter xylinus* used the glucose in the ink structure as a metabolite and produced cellulose pellicles at the surface of printed features, where the oxygen supply was sufficient. The movement and the population of bacteria were observable by staining the live bacteria with green fluorescent dye. The bacteria showed a high population near the surface and high bioactivity producing cellulose layers (Fig. 6b). The CNF concentration and the bioactivity of bacteria in the hydrogel structure were the key factors in determining the thickness of the biosynthesized BC layer. As a result, the thickness of the biosynthesized BC tended to decrease gradually as the amount of CNF increased (Fig. 6c). As the number of bacteria inoculated in the ink increased, the thickness of BC increased slightly.

The inner diameter of the CNF/BC composite tubes was determined by the diameter of CNF ink filaments printed using SMAP. The diameter of a CNF ink filament was controlled by adjusting the printing speed and printing pressure. The diameter

followed an inversely proportional relationship with the printing speed, as shown in Fig. 6d. In contrast, the diameter increased with printing pressure, following a linear relationship (Fig. 6e). The thickness of the tubular structure was determined by the thickness of the BC biosynthesized at the ink filament surface. The bacteria were incubated as printed for up to 14 days and the thickness was measured (Fig. 6f). BC thickness increased linearly in the first 6 days, but no further increases in thickness were observed for the next 8 days.

For preparation of the blood vessel model, a hollow CNF/BC tube was prepared by printing a straight line of CNF hydrogel containing bacteria using SMAP, followed by incubation for 7 days and the subsequent removal of templating CNF hydrogel from the product. The treatment of a hollow CNF/BC tube with $Ca^{2+}$ ion solution (CNF/BC/$Ca^{2+}$) enhanced the mechanical properties of the tube. Collagen, one of the key proteins found in the extracellular matrix, was immobilized to CNF/BC/$Ca^{2+}$ tubes to improve cell adhesion (Fig. 6g). Fourier transform infrared (FT-IR) spectra confirmed the collagen immobilization to BC with amide I and amide II peaks appeared at 1640 and 1545 cm$^{-1}$, respectively (Fig. 6h).

**A blood vessel model of hollow CNF/BC tubes**. To evaluate cell compatibility of the CNF/BC tubes, fibroblast cells were seeded on the inner wall of the tube and the cell growth rate was evaluated with an alamarBlue assay (Fig. 7a). The relative metabolic activity of cells cultured for 7 days showed that the cell growth rate of BC and CNF/BC/$Ca^{2+}$ increased slightly at a similar level, while that of collagen-introduced CNF/BC/$Ca^{2+}$ (CNF/BC/col) increased rapidly. In general, higher cell adhesion capacity was associated with faster formation of tissues, especially due to interactions between the cells and the scaffold. Cell density increased significantly in the case of CNF/BC/col (Fig. 7b–d). The morphology of proliferating cells also showed healthy growth of CNF/BC/col. Cells did not spread properly to the surface of BC and CNF/BC, but instead formed a cluster or restraining features. Meanwhile, cells seeded at the surface of CNF/BC/col spread as observed with the fibroblast in healthy growth by staining the cytoskeleton with 4′,6-diamidino-2-phenylindole (DAPI)/phalloidin (Fig. 7e–g).

A microfluidic vascular system of a single fibroblast cell was prepared with a 3D perfusion chip (Fig. 7h). The cell suspension flowed through the CNF/BC/col inside and was allowed to incubate for cell attachment and proliferation. The cells grew and proliferated at the inner wall of the CNF/BC/col tube (Fig. 7i). Cell spreading at the inner wall of CNF/BC/col was also observed with field-emission scanning electron microscopy (FE-SEM). The cells were attached tightly at the nanofibrous surface, confirming the healthy growth of cells (Fig. 7j).

SMAP enabled simple and diverse CNF/BC structuring by the direct printing of removable template structures. A tubular vessel model was prepared by printing a straight line of CNF hydrogel following the procedure described above. The tubular vessel was connected with silicon tubes and a red solution was injected to verify the flow through the tube lines (Fig. 7k, l). A Y-shaped CNF/BC vessel model was also fabricated using SMAP (Fig. 7m). Two separate flows of colored solutions met at the junction without any leakage of solution or breakage of vessel model tubes (Fig. 7n). Flow through the tubes was investigated using fluorescent microscopy. The 3D tubular structure was well established with the protocol and the flow of liquid containing fluorescent particles through the tube was recorded under ultraviolet illumination, verifying the stable mass flow in the line (Fig. 7o). Particle image velocimetry (PIV) analysis demonstrated the distribution of the velocity at which the fluid moved inside the

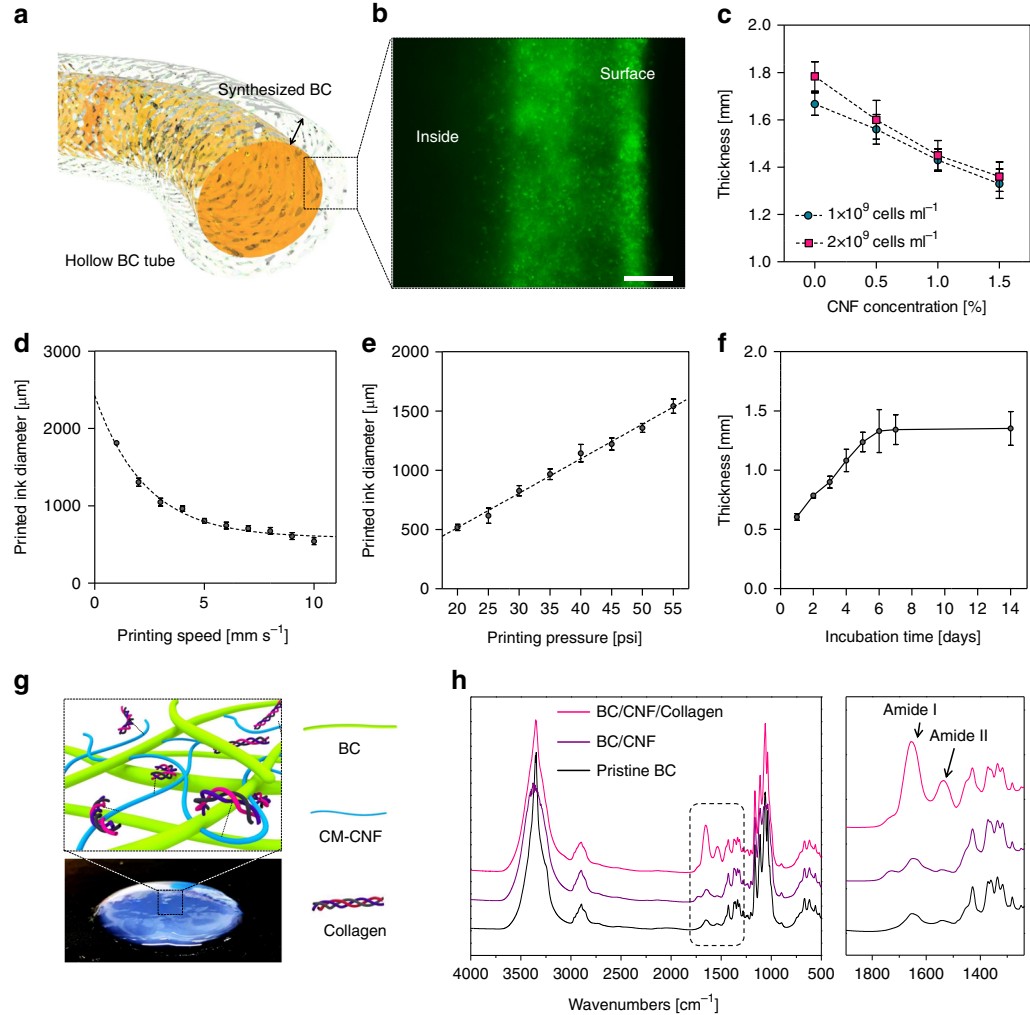

**Fig. 6** Preparation of a hollow cellulose tube. **a** Schematic illustration of a hollow CNF/BC tube. Utilizing the region-specific biosynthesis of BC at the surface of the printed CNF hydrogel, it was possible to fabricate hollow CNF/BC tubes by controlling the printing diameter and incubation time. **b** Movement of bacteria to the surface of the printed CNF/BC hydrogel structure after incubation for 7 days. **c** Change of BC thickness according to CNF content and the number of bacteria in the ink. The diameter of printed CNF/BC hydrogel structures could be controlled through the printing speed of the needle (**d**) and printing pressure applied to an extruding needle (**e**). **f** The thickness of BC biosynthesized at the surface of printed CNF hydrogel structures for different incubation times. **g** Schematic illustration of collagen conjugated to the CNF/BC surface. **h** FT-IR spectra of BC, CNF/BC, and CNF/BC/ collagen. Characteristic peaks for amide I and amide II were observed, confirming the binding of collagen to the CNFs. Scale bar is 50 μm in **b**. Data are presented as mean ± s.d. ($n = 3$ in **c**, $n = 5$ in **f**)

channel (Fig. 7p). Flow velocity was high at the center of the tube and decreased toward the wall. Suspension solution containing fluorescent particles was injected at a rate of $1\,\mathrm{mL\,h^{-1}}$ using an external syringe pump, and the maximum flow rate at the center of the tube was about $75\,\mathrm{mm\,s^{-1}}$ (Fig. 7q). SMAP is a method for the free-form printing of fluidic ink that has not been previously reported, but may be adoptable for the 3D structuring of biopolymers requiring specific conditions such as aeration.

topology and interconnectivity of an oxygen-permeable surface, which was essential for cellulosic biosynthesis of bacteria. The mechanical strength of tubular CNF/BC was enhanced by ionic crosslinking and a biocompatible tube was successfully applied to mass flow mimicking blood streaming and vascular tissue engineering without leakage. A generic tool for the versatile 3D printing of bio-organs and scaffolds may be possible based on the described techniques.

## Discussion

In summary, ink containing active bacteria was prepared using CNFs as rheology modifiers and PTFE microparticles as a highly fluidic and inert matrix. Rapid recovery of the matrix, which was critical to avoiding vertical misprinting, involved filling the needle path with the neighboring matrix particles. PTFE particles with a larger angle of repose represented the high fluidity of the matrix and filled the needle path effectively. The printability of CNF hydrogel was also improved by controlling the printing speed and CNF content in the ink. SMAP enabled the control over the 3D

## Methods

**Preparation of CNF hydrogel**. A CNF hydrogel was produced from kraft pulp (Moorim P&P, Ulsan, Korea). The kraft pulp consisted of 79.4 ± 0.6% cellulose, 18.8 ± 0.2% hemicellulose, and very little amount of lignin and byproducts. The pulp fibers were beaten with a laboratory valley beater for 30 min. Beaten pulp fibers were carboxymethylated, and the following procedure was followed. The wet pulp (dry weight, 70 g) was solvent-exchanged using a series of graded ethanol solutions (50, 70, 90, and 100%) with overhead stirring at 1000 r.p.m. for 20 min. The solvent-exchanged pulp fiber was immersed in a solution of sodium hydroxide (35 g) in isopropanol (3200 mL)/methanol (800 mL) at 65 °C for 30 min. Mono-chloroacetic acid (35 g, Sigma-Aldrich, St. Louis, MO, USA) was added to the cellulose slurry, and the mixture was stirred for 90 min. The carboxymethylated

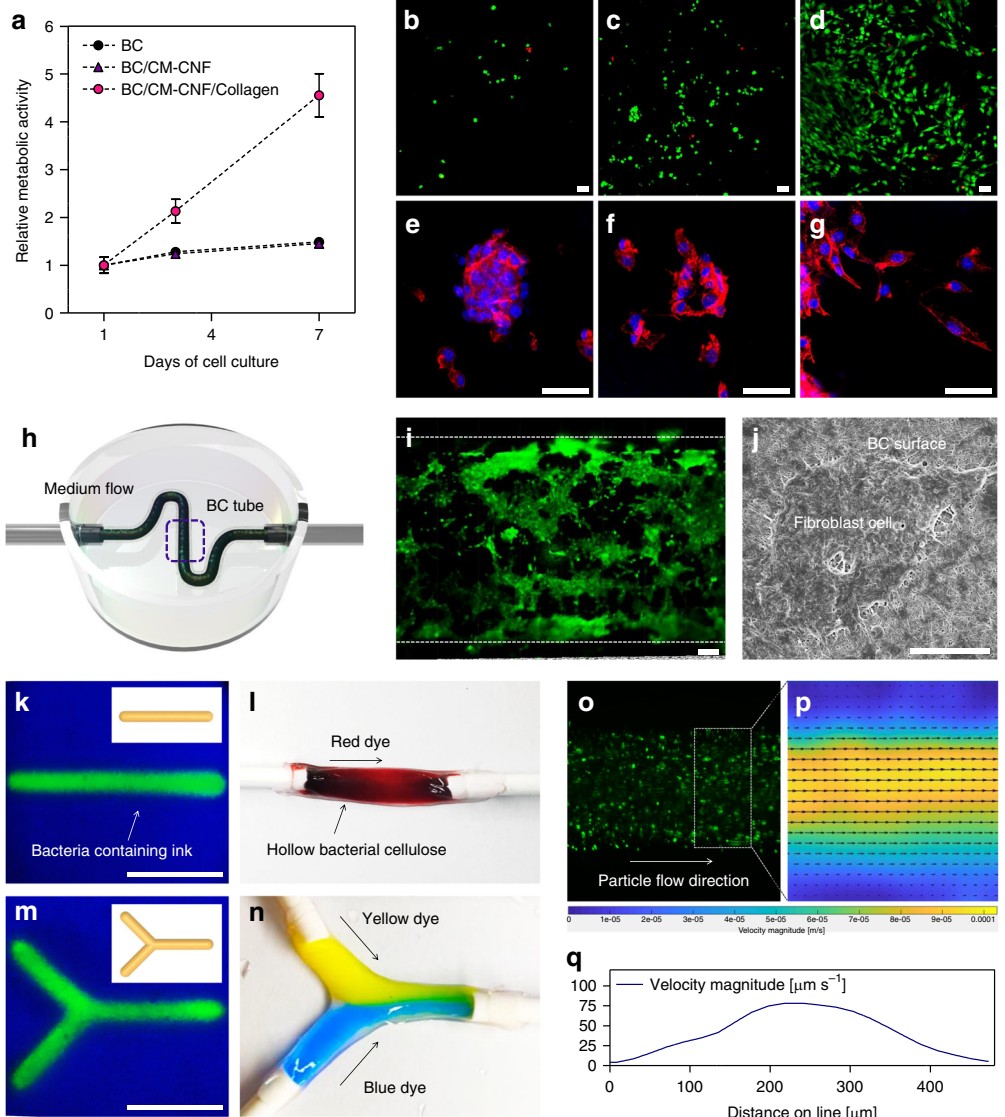

**Fig. 7** A blood vessel model of hollow CNF/BC tubes. **a** Cytocompatibility of CNF/BC tubes using alamarBlue assays. Cell proliferation of tubes was evaluated after incubation for 7 days. Live and dead assay of fibroblast cells cultured for 7 days on BC (**b**), BC/CNF (**c**), and BC/CNF/collagen (**d**) (live cells in green and dead cells in red). Phalloidin/DAPI assay of fibroblast cells cultured for 7 days on BC (**e**), BC/CNF (**f**), and BC/CNF/collagen (**g**). **h** Schematic depicting a vascular CNF/BC tube whose inner wall was incubated with a single fibroblast cell and housed in a 3D perfusion chip. **i**. Fibroblast cells covering vascular CNF/BC tube surfaces were imaged by cross-sectional confocal microscopy. **j** FE-SEM image of a fibroblast cell spreading at the surface of collagen-conjugated CNF/BC tube. **k** A linear CNF/BC tube filled with fluorescent particles. **l** A linear hollow CNF/BC tube was connected to a silicone tube for the flow of red-colored liquid. **m** A Y-shaped hollow CNF/BC tube filled with fluorescent particles. **n** Yellow and blue liquids were injected separately through the Y-shaped tube. **o** The flow of fluorescent particles in the CNF/BC tube. Velocity magnitude fields (**p**) and velocity profiles (**q**) in the hollow CNF/BC tube model. Scale bars are 50 μm in **b**–**g**, 100 μm in **i**, and 10 μm in **j** and 1 cm in **k**, **m**. Data are presented as mean ± s.d. (*n* = 3 in **a**)

cellulose fiber was solvent-exchanged using distilled water and passed through a grinder (Super Masscolloider, Masuko Sangyo Co., Ltd., Japan) to produce the CNF hydrogel. The concentration of the pulp suspension during grinding was 1.5% (w/v). The operation speed and the gap distance between the grinder stones were 1500 r.p.m. and 100 μm, respectively. The number of passes through the grinder was fixed at five.

**Preparation of CNF hydrogel ink containing bacteria**. The concentration of CNF was adjusted from 0 to 1.5%, after which 2.5% (w/w) of mannitol, 0.5% (w/w) of yeast extract, and 0.3% (w/w) of Bacto Peptone were added. The CNF hydrogel was then inoculated with *G. xylinus* (KCCM 40216) obtained from the Korean Culture Center of Microorganisms and stirred for 30 min.

**Measurement of the rheological properties of the CNF ink**. The rheological behavior of the CNF hydrogel ink was characterized using a digital rheometer (MARSIII, Thermo Scientific, Newington, NH, USA) fitted with a parallel plate (35

mm radius) to investigate shear-thinning properties. Prior to rheological characterization, the CNF hydrogel was stirred vigorously and degassed in a centrifuge at 168 × *g*. Various concentrations of CNF hydrogels containing bacteria and nutrients were placed in the plate and subjected to stress-sweep experiments from 0.1 to 1000 Pa at a frequency of 1 Hz and a temperature of 25 °C. The shear viscosity of the CNF hydrogel was obtained at 25 °C, and the measurement was performed in Rot Ramp mode at shear rates from 0.01 to 1000 s$^{-1}$ with a gap size of 1.0 mm. Step-stress measurements were performed at a high-magnitude (300 Pa) and low-magnitude (0.5 Pa) stress.

**Characterization of the PTFE solid matrix**. To determine the solid matrix conditions that promote printing stability, hydrophobic PTFE microparticles (TF 1641, 3M Dyneon, USA) were separated by 100, 200, 300, and 500 microsieves and particle diameters were measured using an optical microscope. For the angle of repose, PTFE microparticles were dropped on a cylinder with a radius of 1 cm and the height was measured. The tangent angle was calculated using the height of the stacked PTFE microparticles and the radius of the cylinder.

**SMAP of CNF hydrogel inks**. The 3D structures of hydrogel ink containing bacteria were printed in a solid matrix of PTFE microparticles using a custom-built 3D printer. The 3D structures were designed using the commercially available Rhinoceros software (Rhinoceros 5.0, Robert McNeel & Associates, Seattle, WA, USA). The designed 3D models were translated into G-code instructions for deposition using slicing software (Cura, Ultimaker, Geldermalsen, The Netherlands). CNF hydrogel ink containing bacteria in a 5 mL syringe was printed in a Petri dish filled with PTFE microparticles. The needle was inserted into the bottom center of the PTFE solid matrix in the Petri dish, and the G-code was sent to the printer using the host software. The ink was extruded through a syringe needle with an inner diameter of 160 μm. The extruding volume and thickness of the lines were controlled by the applied pressure and the printing speed.

**Imaging of morphology and microstructures of the hydrogels**. Composite CNF hydrogels were immersed immediately in liquid nitrogen to maintain porosity. Morphologies of the freeze-dried CNF, BC, and CNF/BC hydrogels were analyzed using a FE-SEM, SUPRA 55VP, Carl Zeiss, Oberkochen, Germany) operating at a voltage of 2 kV. Platinum was sputtered onto the hydrogels at 20 mV for 160 s as a conductive coating for imaging purposes. BC fibers biosynthesized were stained with 5% Calcofluor white solution for 1 h and then washed with deionized water (DW) three times. Fluorescence images of cellulose fibers were collected using a confocal laser microscope (Carl Zeiss, Oberkochen, Germany).

**Measurement of the mechanical properties of the hydrogels**. The viscoelastic properties of the printed hydrogels were measured using the digital rheometer. After printing, the CNF hydrogel ink containing bacteria was incubated for 7 days, immersed in 1% NaOH solution for 24 h, and then washed thoroughly with DW. CNF/BC hydrogels treated with $Ca^{2+}$ (CNF/BC/$Ca^{2+}$) were prepared by immersing CNF/BC hydrogel in a 1% $CaCl_2$ solution for 1 h. To measure the shear modulus, BC, CNF/BC, and CNF/BC/$Ca^{2+}$ hydrogels were cut into circular discs (8 mm in diameter) using a biopsy punch. Oscillatory rheometry in the shear strain-sweep mode was performed at a frequency of 1 Hz. The gel modulus was measured using a parallel plate geometry (8 mm) with a nominal force of 0.2 N and a gap size of 2 mm.

The tensile strength of the hydrogels was investigated using a universal testing machine (UTM, GB/LRX Plus, Lloyd, West Sussex, UK) fitted with a 10 N load cell. For tensile testing, gauge samples 1 cm wide, 2 cm long, and 1.5 mm thick were prepared. Ultimate tensile strength was determined on five wet samples at a speed of 2 mm min$^{-1}$.

**Preparation of a blood vessel model using hollow CNF/BC tubes**. A template for in situ biosynthesis of BC was fabricated by printing straight lines 500 μm in diameter and Y-shaped lines of CNF ink containing the bacteria with SMAP. After incubation for 7 days, the CNF/BC composite hydrogel was washed with DW to remove the CNF ink inside the printed structures, leaving a tubular structure. The prepared CNF/BC tubes were immersed in 1% NaOH solution for 1 day and washed with DW.

The inner wall of the CNF/BC tube was coated with collagen to increase cell adhesion. Collagen dissolved in 0.1 M acetic acid solution was introduced to the CNF/BC tube with a syringe and covalent binding of collagen to the surface was performed using 1-ethyl-3-(3dimethyl aminopropyl)carbodiimide (EDC, Sigma-Aldrich, USA) and N-hydroxysuccinimide (NHS, Sigma-Aldrich, USA) conjugation chemistry. The CNF/BC tube was then immersed in a solution of 0.4 mg mL$^{-1}$ EDC and 0.6 mg mL$^{-1}$ NHS and shaken gently for 30 min. The activated CNF/BC tube was collected and a 1 mg mL$^{-1}$ collagen solution was injected through the tube to form covalent bonds between the $NH_2$ groups of collagen and the COOH groups of CNFs on the inner wall of the CNF/BC hydrogel tube (CNF/BC/collagen). To get rid of EDC and NHS from the CNF/BC tubes, the treated tubes were immersed in 0.1 M $Na_2HPO_4$ for 1 h and washed with DW. The washing process repeated twice and the tubes were subsequently immersed in 1 M NaCl solution for 1 h, followed by rinsing with running DW for 24 h. The chemical structures of the samples were characterized by FT-IR spectroscopy (Nicolet iS5, Thermo Scientific, USA). Thirty-two scans at a resolution of 4 cm$^{-1}$ were made, with a wavenumber range of 4000–600 cm$^{-1}$. Fibroblast cells (ATCC, CRL 1658, at a concentration of $1 \times 10^7$ cells mL$^{-1}$) were seeded on the inner wall of CNF/BC/collagen tube by flowing the cell suspension through the tube and incubating it in a medium consisting of Dulbecco's modified Eagle's medium with 4.5 g L$^{-1}$ glucose, L-glutamine, sodium pyruvate, and 10% fetal bovine serum under humidified 5% $CO_2$ at 37 °C for 1, 4, and 7 days (CNF/BC/cells).

Active perfusion of the vessel model was investigated with CNF/BC cells incubated for 1 day by imaging the flow patterns of the tube. The aqueous suspension of 0.1% (w/v) fluorescent particles flowed through the tube, and the flow field was visualized with PIV. The suspension was injected at a flow rate of 1 mL h$^{-1}$ using a syringe pump, and the images were captured at 10 fps using a fluorescence microscope. A Gaussian filter was applied to each image, and the average background noise was subtracted from the images. The velocity vectors were obtained by applying 2D cross-correlations of the interrogation windows of sizes $128 \times 128$ pixels with 50% overlap for the coarse grid and $64 \times 64$ pixels with 50% overlap for the refined grid system. All image processing procedures were performed in MATLAB.

**Cytotoxicity of composite hydrogels**. A Live and Dead Viability/Cytotoxicity Kit (Invitrogen, Waltham, MA, USA) was used to investigate the viability of the cells. The live and dead assay reagents were added to a Petri dish containing cell spread constructs and incubated for 1 h in the dark. Fluorescence images of the cells were collected using a confocal laser scanning microscope (Carl Zeiss, Germany). To investigate cell proliferation, the composite hydrogels were washed once with phosphate-buffered saline and incubated with alamarBlue (Invitrogen) solution at 37 °C for 4 h. AlamarBlue fluorescence was assayed at 540 nm (excitation) and 590 nm (emission) using a microplate reader (Synergy HT, BioTek). Rhodamine-labeled phalloidin (Alexa Fluor 594, Life Technologies) and DAPI (Sigma, St. Louis, MO, USA) were used according to manufacturer's instructions for F-actin and cell nuclei staining, respectively, to examine cell adhesion on samples. Briefly, cells were fixed with 4% paraformaldehyde for 30 min, permeabilized with 0.1% Triton X-100 for 20 min, and then blocked with 1% bovine serum albumin for 45 min. Rhodamine-labeled phalloidin solution (dilution 1:40) was then added and incubated with the cells at 37 °C for 45 min. The DAPI solution (dilution 1:1000) was then added to the cells and incubated at 37 °C for 45 min.

## Data availability

The data supporting the findings of this work are available in the main text or supplementary materials. Raw data are available from the corresponding author upon reasonable request.

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

## Acknowledgements

We thank Jaehwan Kim at SNU for his help on the BC experiments. J.H. would like to thank Hye Jung Youn for the discussion and comments. This research was supported by the Basic Science Research Program through the National Research Foundation of Korea (NRF) funded by the Ministry of Education (grant number NRF-2018R1D1A1B07049081).

## Author contributions

J.H. conceived the idea and coordinated the research project. S.S. designed and conducted the experiments. D.S. designed the 3D CAD models and modified G. codes. H.K. conducted cell experiments. All authors discussed the results. S.S. and J.H. wrote the paper with input from all authors.

## Competing interests

The authors declare no competing interests.
