## [Peer Review File · Nature Communications]

Reviewers' comments:

Reviewer #1 (Remarks to the Author):

Manuscript # NCOMMS-19-12627 titled "Solid Matrix-Assisted Printing for Three-Dimensional Structuring of a Viscoelastic Medium Surface" relates to additive manufacturing of hydrogels comprising an incubation medium, anionic cellulose nanofibers and cellulose-producing bacteria, thereby achieving biofilms in-situ that conform pre-determined shapes. For this, a printing matrix of PTFE microparticles was used with some level of resolution and dimensional stability. Ionic crosslinking was achieved with Ca²⁺ to enhance the mechanical properties of the 3D cellulose hydrogel structures. Tubular BC structures were obtained and a demonstration is given for the applicability to future biomedical applications.

In general, the work was well conducted and presented. It contains enough elements of novelty to make it deserving of publication. I did not find any flaws but want just to state a few generic aspects for the consideration of the authors:

The novelty statements are not convincing: "In contrast, SMAP allows for flexible design of more complex structures, including spheres, tubes, and connected rings. Especially, BC tubular structures can be used as engineered tissue support for blood vessels or neural regeneration." This is because spheres and tubes have been reported in several manuscripts, dating prior to 2014 – as described in DOI: 10.1002/app.41719 - or after 2014 as described in DOI: 10.1039/C8SM00112J or ref. 21, for spheres, for instance. Furthermore, in ref. 21 complex structures are shown with a resolution of ~50 µm.

The achievement of complex structures has been shown in the references above; furthermore, resolution < 10 µm has been shown in several other reports (for instance, DOI:10.1021/nn5036125). The advantage of the newly introduced 3D-printing as described is therefore not resolution and especially not complexity but the control over the topology and interconnectivity, which is a feature that is achieved uniquely by 3D-printing.

With the method described by the authors, new horizons can be opened to 3D-printing as a versatile fabrication methodology. The analogy with templating as described in ref. 21 is necessary but the disadvantages of 3D-printing (lower resolution) should be clearly highlighted as well as the advantages (higher versatility in topology).

The novelty statement also undermines the other positive aspects of the work by focusing on the printing of bacteria specifically. 3D-printing in powder, as thoroughly described in the paper, is applicable to a wide variety of contexts. For instance, it would substantially expand the many examples described in ref. 19.

The recent work that proposed the use of hydrophobic particles for formation of biofilms of given geometries, ref. 21, is related closely to the present effort. In relation to this prior efforts, the authors indicate that "When 3D structures of BC were prepared in a mold with hydrophobic particles on the wall, only simple structures were available, and it was difficult to separate complex BC structures from the molds." (lines 45-47). These statements are not correct: (1) first, complex structures have been in fact demonstrated as indicated in Ref. 21 and followup work. They include capsule-in-capsule configurations, seamless hollow objects with complex curvatures and open films following shapes as complex as those of an ear. (2) Secondly, in such work, demolding has not been an issue.

So, the authors need to put their work in context, accurately. The main issue is to compare templating vs 3D printing. Templating has a much higher resolution but is limited in terms of topology. This should be more clearly stated

A minor issue in Line 30: "BC is a polysaccharide synthesized from *Gluconacetobacter xylinus*". Please note that many different microorganism strains also produce BC. IN passing, please note that authors refer to *Acetobacter Xylinus* in Fig. 1.

Reviewer #2 (Remarks to the Author):

The authors reported a solid matrix-assisted 3D printing (SMAP) processing technique of an incubation medium surface for the 3D fabrication of bacterial cellulose hydrogel-structures. This technique would allow the preparation of tailored bacterial cellulose-based 3D structures. Some points should be addressed before publication:

There is little information regarding the incubation time needed to produce these structures. Most samples were incubated for 7 days. In the case of the tubular structure, the thickness was evaluated and it is clear that after 8 days no more thickness can be achieved. However, the blood vessel sample was incubated for 6 days. In static conditions, BC is often incubated for 7-14 days. Please, add some comments regarding why the authors used 7 days for most samples and if the incubation time could be controlled by modifying the glucose content of the ink.

The authors should provide information regarding the properties of the BC network (diameter and longitude of BC fibers, aspect ratio of BC fibers, mesh size of BC network). The BC network properties depend on the growing media and conditions. For instance, the BC diameter should be compared with reported diameters of BC-based structures prepared with other techniques. The authors should report if BC network properties (diameter, mesh size, etc.) are constant or depend on the structure they print.

Regarding the rheological characterization of the BC-based hydrogels, the authors only presented strain sweeps. Why? Did the authors performed temperature and frequency sweeps?

Please report the thickness of the samples used for tensile tests.

Fig. 2b. Please indicate the symbols used for the storage modulus (G') and for the loss modulus (G'').

Fig. 2e should be next (not below) to Fig. 2d

Omar Troncoso

Reviewer #3 (Remarks to the Author):

This paper presents 3D fabrication of CM-CNF/BC hydrogels using in situ BC production of bacteria through the PTFE microbeads system, through which air can be supplied to the viscous CM-CNF/bacteria mixtures. The idea of producing the hydrogels are unique and interesting, and the manuscript has a high originality. However, technical repeatability by readers is not sufficient also in terms of science, which should be improved before publication. Details are as follows.

1) The local amount of BC production in the hydrogels may significantly be influenced by the air-feeding levels. The even air-feeding is required to prepare homogeneous hydrogel thicknesses or amounts in throughout the hydrogel parts. How the authors controlled the homogeneous air-feeding in the whole interfaces?

2) The information of the CM-CNF prepared in this study is insufficient. More detailed information should be addressed in Supporting Information, such as the yield of CM-pulp, the degree of substitution, molecular weights, the degree of nanofibrillation, etc., otherwise readers cannot repeat the experiment. The details of kraft (not craft) pulp also should be addressed, such as the hemicellulose content, degree of polymerization, etc. In general, dissolving pulps with high cellulose contents are used in carboxymethylation. However, in this study, the kraft pulp containing a

significant amount of hemicelluloses was used. Please explain the reason.

3) In this study, amidation was used to fix collagen to BC. Were the EDC, NHS used in this study are really safe to be used as artificial blood vessels? The amidation used in this system is not simple, and some residual EDC or NHS may remain in the hydrogels as counterions of carboxy groups of CM-CNF.

4) How the authors removed the BC cells? The 0.1% NaOH post-treatment may have partly removed CM-CNF components.

Reviewer #4 (Remarks to the Author):

The reported work attempts to mainly propose preparation of 3D ink containing active bacteria. The attempt itself is quite interesting. The reviewer will refer here on the side of the bacterium, *G. xylinus* as a component of the ink. In this ink, behavior of *G. xylinus* would be critical. The authors tried to control it by the oxygen supply, although the bacterium is quite sensitive to the environmental change (stress). In this point, there are several critical papers that the authors should need to cite; for example, Nagashima et al (2016), Czaja et al. (2004), Apelgren et al. (2019).

By considering the sensitivity of the bacterium to the culture condition and thereby feasibility of its changing behavior, the reviewer wonder if the obtained product could exhibit the stabilized quality. Therefore, the authors should be more careful in observing and realizing the bacterium behavior in the ink, not simply due to dependence on the oxygen supply.

Response to reviewer's comments

Reviewer #1

Q1. The advantage of the newly introduced 3D-printing as described is not resolution and especially not complexity but the control over the topology and interconnectivity, which is a feature that is achieved uniquely by 3D-printing.

Response: We fully agree with the comment. The phrase “high resolutions” has been removed in the revised manuscript (MS). The MS has been modified to emphasize the features of SMAP as the reviewer mentioned. In addition, different designs for the 3D BC structures have been included in the revised MS. To clarify this point, the following text has been included:

“The fabrication of a versatile free-form structure of bacterial cellulose (BC) has been proven impossible due to restricted oxygen supplies at the medium and the dimensional instability of hydrogel printing.”

“Complex 3D structures of BC with a resolution of about 50 μm were prepared in a mold with hydrophobic particles. Higher resolutions of the BC could be fabricated in patterned superhydrophobic–hydrophilic surfaces and at the interface of a soft-lithographic PDMS mold. Because of the limited choice of needles for direct 3D printing, the resolution of SMAP using a viscous ink is comparatively low. However, SMAP enables control over the topology and interconnectivity, which is a feature that is achieved uniquely by 3D printing.”

“Successfully constructed complex structures included coil, tetrahedron, connected ring, stacked lattice, and sandglass structures, which have not been reported in 3D printing to date (Fig. 1d).”

Q2. The disadvantages of 3D-printing (lower resolution) should be clearly highlighted as well as the advantages (higher versatility in topology).

Response: This point has been included in the revised MS. The following text has been included:

“Complex 3D structures of BC with a resolution of about 50 μm were prepared in a mold with hydrophobic particles. Higher resolutions of the BC could be fabricated in patterned superhydrophobic–hydrophilic surfaces and at the interface of a soft-lithographic PDMS mold. Because of the limited choice of needles for direct 3D printing, the resolution of SMAP using a viscous ink is comparatively low. However, SMAP enables control over the topology and interconnectivity, which is a feature that is achieved uniquely by 3D printing.”

Q3. The novelty statement also undermines the other positive aspects of the work by focusing on the printing of bacteria specifically. 3D-printing in powder, as thoroughly described in the paper, is applicable to a wide variety of contexts. For instance, it would substantially expand the many examples described in ref. 19.

Response: We appreciate this comment. As the reviewer mentioned, SMAP is applicable in a variety of contexts. Here, we focused on bacterial cellulose, but SMAP can be expanded to the structures of other living inks. Furthermore, different types of solid particles, including metal, polymer, ceramic and wood, can be used as a solid matrix. Unfortunately, it is not possible to provide relevant experimental results, but this point has been included in the revised MS to inform the reader of the wide applicability of SMAP. To clarify this point, the following text has been included:

“In contrast, SMAP allows for flexible design of more complex structures, including spheres, tubes, coils and connected rings. Especially, BC tubular structures can be used as

engineered tissue support for blood vessels or neural regeneration. SMAP can be also applied to structure hydrogel inks containing various types of aerobic microorganisms. In addition, it is expected that SMAP will be expanded by diversifying the solid particles constituting the matrix such as metals, ceramics, and wood materials, which would upgrade the conventional 3D printing technology further.”

Q4. The recent work that proposed the use of hydrophobic particles for formation of biofilms of given geometries, ref. 21, is related closely to the present effort. In relation to this prior efforts, the authors indicate that “When 3D structures of BC were prepared in a mold with hydrophobic particles on the wall, only simple structures were available, and it was difficult to separate complex BC structures from the molds.” (lines 45-47). These statements are not correct: (1) first, complex structures have been in fact demonstrated as indicated in Ref. 21 and followup work. They include capsule-in-capsule configurations, seamless hollow objects with complex curvatures and open films following shapes as complex as those of an ear. (2) Secondly, in such work, demolding has not been an issue.

Response: We completely agree with the reviewer’s comment. The statements have been corrected and modified in the revised MS. We focused on the higher versatility of the topology as a feature of SMAP in the revised MS. To clarify this point, the following text has been included:

“Complex 3D structures of BC with a resolution of about 50 μm were prepared in a mold with hydrophobic particles. Higher resolutions of the BC could be fabricated in patterned superhydrophobic–hydrophilic surfaces and at the interface of a soft-lithographic PDMS mold. Because of the limited choice of needles for direct 3D printing, the resolution of SMAP using a viscous ink is comparatively low. However, SMAP enables control over the topology and interconnectivity, which is a feature that is achieved uniquely by 3D printing.”

Q5. A minor issue in Line 30: “BC is a polysaccharide synthesized from Gluconacetobacter xylinus”. Please note that many different microorganism strains also produce BC. IN passing, please note that authors refer to Acetobacter Xylinus in Fig. 1.

Response: We appreciate this comment. This error has been corrected in the revised MS.

“BC is a polysaccharide synthesized from many different microorganism strains and a hydrogel of a complex networked structure in which nanometer-sized cellulose fibers are intertwined.”

Reviewer #2:

Q1. There is little information regarding the incubation time needed to produce these structures. Most samples were incubated for 7 days. In the case of the tubular structure, the thickness was evaluated and it is clear that after 8 days no more thickness can be achieved. However, the blood vessel sample was incubated for 6 days. In static conditions, BC is often incubated for 7-14 days. Please, add some comments regarding why the authors used 7 days for most samples and if the incubation time could be controlled by modifying the glucose content of the ink.

Response: We appreciate the comments of the reviewer. In the case of the blood vessel sample, it was cultured for 7 days. This information has been corrected in the revised manuscript (MS).

Detailed information regarding BC production according to the incubation conditions, including glucose concentration, CNF concentration, and printing depth, have been included in the revised MS.

The following results are presented in Fig. 4 in the revised MS.

“For preparation of the blood vessel model, a hollow CNF/BC tube was prepared by printing a straight line of CNF hydrogel containing bacteria using SMAP, followed by incubation for 7 days and the subsequent removal of templating CNF hydrogel from the product.”

“The network structure of BC changes depending on the growing media conditions. The yield, fiber diameter, and mesh size of BC were analyzed by varying the concentration of mannitol, which was a carbon source contained in the ink, to 1.25, 2.5, 5, and 10%.....A straight line and the curved edge of BC structure produced at the same printing depth were characterized, and no significant differences in fiber diameter or mesh size of the BC were observed, implying the homogeneous BC hydrogels exhibited morphological diversity (Fig. S2b, c).”

Q2. The authors should provide information regarding the properties of the BC network (diameter and longitude of BC fibers, aspect ratio of BC fibers, mesh size of BC network). The BC network properties depend on the growing media and conditions. For instance, the BC diameter should be compared with reported diameters of BC-based structures prepared with other techniques.

Response: We appreciate this comment. Unfortunately, it was not possible to provide the length or aspect ratio of the BC fibers because it is extremely difficult to take images of the full length of individual fibers. However, we analyzed the properties of the BC network, including the fiber diameter and mesh size, using FE-SEM images; this information has been included in the revised MS. The average diameters of BC fibers were found to be almost constant at about 55 nm. The mesh size of the BC was analyzed in accordance with the reference (Grande, Cristian J., et al., "Morphological characterization of bacterial

cellulose-starch nanocomposites", *Polymers and Polymer Composites* 16.3 (2008): 181-185). In that paper, the mesh size of BC was defined as the distance between junction points. The diameters of the BC fibers appeared to be similar to those in previously reported studies. However, the mesh size varied according to the addition of CNF compared with the BC obtained using the static culture. We believe that the restricted cell locomotion in the CNF containing culture medium reduced the formation of BC. These results have been included in the revised MS.

Additionally, Fig. S2 has been included in the revised MS as follows:

“The network structure of BC changes depending on the growing media conditions. The yield, fiber diameter, and mesh size of BC were analyzed by varying the concentration of mannitol, which was a carbon source contained in the ink, to 1.25, 2.5, 5, and 10%.....A straight line and the curved edge of BC structure produced at the same printing depth were characterized, and no significant differences in fiber diameter or mesh size of the BC were observed, implying the homogeneous BC hydrogels exhibited morphological diversity (Fig. S2b, c).”

Q3. The authors should report if BC network properties (diameter, mesh size, etc.) are constant or depend on the structure they print.

Response: Thank you for your comment. We printed various structures and analyzed the network properties of BC using FE-SEM images. As shown in Fig. S2, it was confirmed that there were no significant changes in the fiber diameter or mesh size of BC. This is now described in the revised MS, and the following results are included in the Supporting information.

“Possible changes in the network structure of the BC according to the printed structure

were investigated using two extreme shapes of BC (Fig. S2a). A straight line and the curved edge of BC structure produced at the same printing depth were characterized, and no significant differences in fiber diameter or mesh size of the BC were observed, implying the homogeneous BC hydrogels exhibited morphological diversity (Fig. S2b, c).”

Q4. Regarding the rheological characterization of the BC-based hydrogels, the authors only presented strain sweeps. Why? Did the authors performed temperature and frequency sweeps?

Response: As the reviewer requested, we performed a frequency sweep to set the frequency for strain sweeping on the BC hydrogel. This information is included in the Supporting information. The change in shear modulus of the BC hydrogel was measured as the temperature was increased from 4 °C to 80 °C. The BC showed no significant change in shear modulus at all temperature conditions. This information is included in the Supporting information.

“Frequency sweep tests are widely used to obtain information regarding the stability of three-dimensional cross-linked networks. After the strain sweep test of Ca²⁺ ions treated CNF/BC composites, the condition for the frequency sweeps was selected at 0.5% strain to ensure the linear viscoelastic range during the test (Fig. S5a). G' and G'' values of Ca²⁺ ions treated CNF/BC composites were constant in the temperature range from 4 °C to 80 °C confirming the high thermal stability of the hydrogel (Fig. S5b).”

Q5. Please report the thickness of the samples used for tensile tests.

Response: This information is included in the revised MS.

“For tensile testing, gauge samples 1 cm wide, 2 cm long and 1.5 mm thick were prepared.”

Q6. Fig. 2b. Please indicate the symbols used for the storage modulus (G') and for the loss modulus (G'').

Response: This is corrected in the revised MS.

Q7. Fig. 2e should be next (not below) to Fig. 2d

Response: This is corrected in the revised MS.

Reviewer #3:

Q1. The local amount of BC production in the hydrogels may significantly be influenced by the air-feeding levels. The even air-feeding is required to prepare homogeneous hydrogel thicknesses or amounts in throughout the hydrogel parts. How the authors controlled the homogeneous air-feeding in the whole interfaces?

Response: We agree with the reviewer's comment. As the reviewer mentioned, the supply of homogeneous air-feeding was one of the critical factors for the production of BC. The same volume of ink containing bacteria was printed in the solid matrix at a different printing depth. After incubation for 7 days, the weight, diameter and network density of the BC were measured. No significant differences in yield, diameter, or network density of the BC were observed, and it was determined that the solid matrix supplied even air-feeding in the printing environment.

“The production of BC in hydrogels may significantly be influenced by the air-feeding levels. It is critical to confirm even BC production with respect to the printing depth for the fabrication of homogeneous BC hydrogel structures. Interestingly, the mesh sizes were also similar for different printing depths, which differs from the results obtained for

different medium conditions (Fig. 4k, l).”

Q2. The information of the CM-CNF prepared in this study is insufficient. More detailed information should be addressed in Supporting Information, such as the yield of CM-pulp, the degree of substitution, molecular weights, the degree of nanofibrillation, etc., otherwise readers cannot repeat the experiment. The details of kraft (not craft) pulp also should be addressed, such as the hemicellulose content, degree of polymerization, etc. In general, dissolving pulps with high cellulose contents are used in carboxymethylation. However, in this study, the kraft pulp containing a significant amount of hemicelluloses was used. Please explain the reason.

Response: We agree with the reviewer’s comments. More information regarding the CM-CNF and kraft pulp has been included in the revised MS. The kraft pulp was used to improve cost effectiveness and mass production in the future. To ensure reproducibility of the experiment, we agree that information on CM-CNF is important. The degree of substitution of CM-CNF was 1.14mmol/g and the yield was 80% based on the amount of CM-pulp obtained after the reaction. The degree of nanofibrillation was confirmed by AFM and the fiber diameter was measured to be 18.6 nm. The kraft pulp consisted of 79.4% ± 0.6% cellulose, 18.8% ± 0.2% hemicellulose and very small amounts of lignin and byproducts. The chemical composition of the pulp fiber was measured according to a TAPPI method (T 203 om-93). There was a paper that once-dried pulp with higher hemicellulose content that was fibrillated into 10–20 nm wide fibers as easily as the never-dried pulp, while the once-dried pulp with lower hemicellulose content was not fibrillated into uniform nanosized fibers. (Iwamoto, Shinichiro, Kentaro Abe, and Hiroyuki Yano. "The effect of hemicelluloses on wood pulp nanofibrillation and nanofiber network characteristics." *Biomacromolecules* 9.3 (2008): 1022-1026.). High-hemicellulose-content

pulps are favorable for the preparation of MFC, meaning that pulps with high hemicellulose content are beneficial for homogeneous carboxymethylation. (Siró, Istvan, et al. "Highly transparent films from carboxymethylated microfibrillated cellulose: the effect of multiple homogenization steps on key properties." *Journal of Applied Polymer Science* 119.5 (2011): 2652-2660.). We also used pulp with a high hemicellulose content to promote uniform and efficient nanofiber formation.

“The CNF was highly nanofibrillated and the diameter of CNF was about 18.6 nm, as measured with atomic force microscopy, which was much smaller than the diameter of BC fibers of about 55 nm; this is because the CNFs did not have a tight junction between the fibers, unlike BC fibers (Fig. S3).”

“Fig. S3 Degree of nanofibrillation of CM-CNF. a. AFM image of CM-CNF. b. Fiber diameter distribution of CM-CNF. The average fiber diameter was 18.6 nm. c. Conductivity titration curves of pristine pulp cellulose and carboxymethylated pulp cellulose. The degree of substitution was 1.14 mmol/g and the yield of CM-pulp was 80%. The viscosity average molecular weight of CM-CNF was 70,000. d. FTIR spectra of pulp cellulose before and after carboxymethylation. The scale bar is 500 nm in (a).”

“The kraft pulp we used consists of $79.4\% \pm 0.6\%$ cellulose, $18.8\% \pm 0.2\%$ hemicellulose, and small amounts of lignin and byproducts.”

Q3. In this study, amidation was used to fix collagen to BC. Were the EDC, NHS used in this study are really safe to be used as artificial blood vessels? The amidation used in this system is not simple, and some residual EDC or NHS may remain in the hydrogels as counterions of carboxy groups of CM-CNF.

Response: We appreciate the comment of the reviewer. We used EDC/NHS chemistry to

conjugate collagen to the BC/CM-CNF complex. Bioconjugation using EDC and NHS has been admitted generally as a non-cytotoxic and biocompatible chemistry. (Pieper, J. S., et al. "Development of tailor-made collagen–glycosaminoglycan matrices: EDC/NHS crosslinking, and ultrastructural aspects." *Biomaterials* 21.6 (2000): 581-593.), (Pieper, J. S., et al. "Preparation and characterization of porous crosslinked collagenous matrices containing bioavailable chondroitin sulphate." *Biomaterials* 20.9 (1999): 847-858.). Regarding the reviewer's concerns, positively charged EDC might remain at the surface of CM-CNF as a counterion. To inactivate and remove EDC and NHS, the samples were carefully washed with 0.1 M Na₂HPO₄, 1M NaCl and DI water. The detailed washing process has been included in the experimental section of the revised manuscript (MS).

“To inactivate and get rid of EDC and NHS from the CNF/BC tubes, the treated tubes were immersed in 0.1 M Na₂HPO₄ for 1 h and washed with DW. The washing process repeated twice and the tubes were subsequently immersed in 1M NaCl solution for 1 h followed by rinsing with running DW for 24 h.”

Q4. How the authors removed the BC cells? The 0.1% NaOH post-treatment may have partly removed CM-CNF components.

Response: Thank you for the comment. This was a typo. The correct concentration is 1.0% NaOH, as mentioned in Measurement of the Mechanical Properties of the Hydrogels. It has been corrected in the revised MS.

Reviewer #4:

Q1. The authors tried to control it by the oxygen supply, although the bacterium is quite sensitive to the environmental change (stress). In this point, there are several critical

papers that the authors should need to cite; for example, Nagashima et al (2016), Czaja et al. (2004), Apelgren et al. (2019).

Response: We appreciate your comment. The following references have been included in the revised MS:

“Viscous inks containing biocompatible rheology modifiers have been used to form a variety of features on substrates via direct 3D printing. (Apelgren et al., 2019).”

“The network structure of BC changes depending on growing media conditions. (Nagashima et al., 2016, Czaja et al., 2004)”

Q2. By considering the sensitivity of the bacterium to the culture condition and thereby feasibility of its changing behavior, the reviewer wonder if the obtained product could exhibit the stabilized quality. Therefore, the authors should be more careful in observing and realizing the bacterium behavior in the ink, not simply due to dependence on the oxygen supply.

Response: We agree with this comment. We further investigated the stabilized quality of BC production. The yield, fiber diameter, and network density of BC were characterized according to the medium conditions, including the concentrations of glucose and CNF, as well as the printing depth of the solid matrix and printing structures. This information has been included in the revised MS.

“The network structure of BC changes depending on the growing media conditions. The yield, fiber diameter, and mesh size of BC were analyzed by varying the concentration of mannitol, which was a carbon source contained in the ink, to 1.25, 2.5, 5, and 10%.....A straight line and the curved edge of BC structure produced at the same printing depth were characterized, and no significant differences in fiber diameter or mesh size of the BC were

observed, implying the homogeneous BC hydrogels exhibited morphological diversity (Fig. S2b, c).”

REVIEWERS' COMMENTS:

Reviewer #1 (Remarks to the Author):

I have read the revised version of the manuscript and the rebuttal letter. The authors have addressed satisfactorily all of the questions I raised. Likewise, I observe that the same may apply to the other reviewers. I consider that the revised manuscript "Solid Matrix-Assisted Printing for Three-Dimensional Structuring of a Viscoelastic Medium Surface" is now suitable for publication.

Reviewer #2 (Remarks to the Author):

The manuscript is ready for publication.

Reviewer #3 (Remarks to the Author):

Because the authors are carefully considered my comments and revised reasonably. Therefore, I think this paper is acceptable for publication in Nature Communications.

Reviewer #4 (Remarks to the Author):

The authors have answered sincerely to my comments, and therefore, as the reviewer #4, I would tell you that my concerns were cleared by their answer.
Thus, I am happy to say that it would be accepted from my points of views.